# Sphingosine-1-Phosphate: Its Pharmacological Regulation and the Treatment of Multiple Sclerosis: A Review Article

**DOI:** 10.3390/biomedicines8070227

**Published:** 2020-07-18

**Authors:** Stanley Cohan, Elisabeth Lucassen, Kyle Smoot, Justine Brink, Chiayi Chen

**Affiliations:** Providence Multiple Sclerosis Center, Providence Brain and Spine Institute, Providence St, Vincent Medical Center, Portland, OR 97225, USA; Stanley.Cohan@providence.org (S.C.); elisabeth.lucassen@providence.org (E.L.); Kyle.smoot@providence.org (K.S.); Justine.Brink@providence.org (J.B.)

**Keywords:** multiple sclerosis, sphingosine-1-phosphate modulators

## Abstract

Sphingosine-1-phosphate (S1P), via its G-protein-coupled receptors, is a signaling molecule with important regulatory properties on numerous, widely varied cell types. Five S1P receptors (S1PR1-5) have been identified, each with effects determined by their unique G-protein-driven downstream pathways. The discovery that lymphocyte egress from peripheral lymphoid organs is promoted by S1P via S1PR-1 stimulation led to the development of pharmacological agents which are S1PR antagonists. These agents promote lymphocyte sequestration and reduce lymphocyte-driven inflammatory damage of the central nervous system (CNS) in animal models, encouraging their examination of efficacy in the treatment of multiple sclerosis (MS). Preclinical research has also demonstrated direct protective effects of S1PR antagonists within the CNS, by modulation of S1PRs, particularly S1PR-1 and S1PR-5, and possibly S1PR-2, independent of effects upon lymphocytes. Three of these agents, fingolimod, siponimod and ozanimod have been approved, and ponesimod has been submitted for regulatory approval. In patients with MS, these agents reduce relapse risk, sustained disability progression, magnetic resonance imaging markers of disease activity, and whole brain and/or cortical and deep gray matter atrophy. Future opportunities in the development of more selective and intracellular S1PR-driven downstream pathway modulators may expand the breadth of agents to treat MS.

## 1. Introduction

Although it was long thought that sphingosine, largely generated from hydrolysis of ceramides within lysosomes [1,2], served primarily as a component of the structural sphingolipid family of molecules, it is now recognized for its importance as a component of sphingosine-1-phosphate (S1P), a crucial messenger molecule. S1P is generated by sphingosine kinase 1 or 2 (sk1, sk2), primarily in red blood cells, platelets and endothelial cells [3,4] (Figure 1). Their G-protein-coupled receptors are widely distributed in many organs and tissues, on cell surface plasma membranes, and on the endoplasmic reticulum and cell nuclei, but it is cytoplasmic membrane-bound receptor sites that have attracted the greatest attention [4]. Five S1P receptor (S1PR-1-5) subtypes have been identified [5,6] (Figure 2a). The translational intracellular pathways activated by S1P-receptor (S1PR) interaction are highly varied because of the multiplicity of receptor subtypes, the varied G-proteins to which they are coupled, the multiple downstream pathways linked to G proteins, and the wide variety cells which express S1PRs (Figure 2b; Table 1). Investigations into the physiological impact of S1P and its receptors, as well as its potential contribution to disease pathogenesis, have revealed important opportunities for therapeutic interventions. This is particularly the case in the treatment of multiple sclerosis (MS). In the past 10 years, three disease modifying therapies (DMT)s which modulate S1PRs, fingolimod (FGM), siponimod (SPM), and ozanimod (OZM), have been approved for treatment of MS, and a fourth agent, ponesimod (PNM) is under regulatory review.

The first part of this article describes the results of preclinical research, which has expanded the knowledge of the physiological roles of S1P and its receptors, and a discussion of the specific mechanisms of action of the S1PR modulators. This work did not only reveal important insights into S1PR-based DMT mechanisms of action, but also widened the perspective on future therapeutic opportunities for this class of molecules. The second part of this chapter summarizes the results of the pivotal clinical trials of FGM, SPM, OZM, and the phase II and phase III trials of PNM upon clinical efficacy, as well as the reported adverse events (AE) of these S1PR modulators.

Fingolimod (FGM) was the first S1PR modulator to be approved for the treatment of relapsing forms of MS (RMS). Preclinical and clinical studies demonstrated that its use results in the sequestration of lymphocytes, particularly central memory and naïve T and B cell lymphocytes in peripheral lymphoid organs [7,8,9,10,11]. Pathological studies also demonstrated reduced pro-inflammatory cells within the CNS following treatment with this class of agents, and it had been widely believed that the sequestration of these potentially pro-inflammatory cells within lymphoid organs accounted for their therapeutic efficacy in MS [7,8,9,10]. However, preclinical investigations, further detailed below, strongly suggest that these agents may also act directly within the CNS, where there is an abundance of S1PR expression, to ameliorate the impact of inflammatory disease.

## 2. Preclinical Studies

As noted above, S1P is a major intracellular signaling molecule via its G-protein-coupled membrane bound receptors [12,13,14,15]. Because the S1PR subsets are coupled to different G-proteins, they can modulate multiple downstream intracellular pathways (Table 1). 

Although S1P is critically essential for normal CNS development and maturation [3,4,16,17] and may regulate synaptic function [18], it may also be cytotoxic at elevated concentrations, such as when there is a genetic deficiency in its degradative enzymes, producing neuronal apoptosis [19,20,21,22,23,24]. S1P also regulates calcium signaling [25], and may promote presynaptic calcium overload and cell death [26].

It is now known that S1P, through its interaction with S1PR-1, expressed on the surface of CCR7+ naïve, central memory B, and T cell lymphocytes, regulates the trafficking of these cells from peripheral lymphoid organs [27,28,29,30]. The S1PR-1-expressing lymphocytes egress in response to the S1P gradient (Figure 3). This effect of S1P makes its metabolic and translational pathways attractive potential therapeutic targets for the treatment of cell-mediated immunologic disorders such as MS. 

### 2.1. S1P Receptors

Fingolimod, approved for the treatment of relapsing forms of MS, is a non-selective S1PR modulator, which once phosphorylated (FGM-P) by sk2 [31], has affinity for S1PRs 1, 3, 4, and 5 [7,31]. The interaction of FGM-P with the S1PR-1 on the surface of CCR7+ lymphocytes results in the internalization and degradation of S1PR-1 [32,33] (Figure 4), promoting the sequestration of naïve, central memory T, and central memory B cell lymphocytes in peripheral lymphoid organs [34]. The circulation of central effector B and T cells, which do not express CCR7, are unaffected by FGM-P. When used in patients with relapsing forms of MS, FGM results in robust clinical and MRI evidence of efficacy, in parallel with marked reduction in circulating B and T cell numbers. 

It has been well-established that the CNS also expresses S1PRs, on neurons, astrocytes, microglia, and oligodendroglia (OLG) during development, maturation, and adult cell states [35,36,37]. Furthermore, FGM-P, and the newer S1PR modulators SPM, OZM, and PNM, readily cross the blood brain barrier (BBB), and are selectively accumulated within the CNS. Unlike FGM, these latter 3 agents do not require prior phosphorylation to be pharmacologically active. It is possible that some of the therapeutic benefits of S1PR modulators result directly from their effects upon CNS receptors, and in the discussion that follows, we present evidence that supports such a role for these agents. Furthermore, there is evidence to suggest that the therapeutic effects of these agents depends, at least in part, upon their interaction with CNS S1PRs [38].

The use of whole animal inflammatory CNS models, such as experimental allergic encephalitis (EAE), non-inflammatory models of CNS demyelination using in situ cuprizone or lyolecithin, in whole animal, brain tissue slice, and CNS cell culture models [39,40,41,42] have all provided important insights into fundamental S1P and S1PR physiology. They have also expanded knowledge of the mechanisms by which S1PR modulation regulates CNS development [4,17], and may protect the CNS during inflammatory insult [38]. What follows is a description of preclinical research which has provided important insights into the role of selective S1PR subtypes, the effects of their modulation/inhibition, and also the use of selective S1PR agonists and antagonists to further clarify the physiological roles of these receptors, and their impact on inflammatory CNS insults. 

Although multiple S1PRs are expressed in the mammalian CNS, S1PR-1, 2 and 5 have attracted the most interest because of the pivotal roles they play in the development and myelinating function of OLGs, the regulation of astrocytes and microglia, and in maintaining the integrity of the BBB.

#### 2.1.1. S1PR-1

During CNS inflammation, both S1PR-1 and S1PR-3 are up-regulated by astrocytes and are associated with astrocyte activation and increased production of glial acidic fibrillary protein (GFAP) [3]. The up-regulation of S1PR-1 is seen on activated GFAP-expressing astrocytes in or near active MS lesions [43,44,45]. During active inflammation there is also S1PR-1-dependent up-regulation of microglia (MGL) that is enhanced in the presence of S1P to further increase inflammation [46]. By contrast, in EAE produced in astrocyte S1PR-1 knockdown mice, there is an inhibition of astrocyte activation [38]. Furthermore, selective S1PR-1 blockade enhances maturation of OLGs [40]. Thus S1PR-1 blockade is a potentially important pharmacological target to reduce astrogliosis and promote remyelination in MS. Although S1PR3 up-regulation occurs in activated astrocytes, the role of S1PR-3 in producing significant cardiovascular effects, via its regulation of the KACh potassium channel [47,48,49,50], has diminished interest in it as a target for modulation in MS.

#### 2.1.2. S1PR-2

S1PR-2 is expressed on oligodendroglial progenitor cells (OPC) and macrophages (MPG) [51,52]. In mice with EAE, S1PR-2 activation promotes breakdown of BBB, which is prevented by the use of selective endothelial cell S1PR-2 blocking agents [53], and in endothelial S1PR-2 knockout EAE mice. In these latter models there is also an inhibition of fibrinogen extravasation into the CNS, as well as the inhibition of MPG and MGL recruitment into CNS lesions [41]. To eliminate the potentially confounding effects of systemic inflammation in EAE, models employing in situ injection of lyolecithin-induced demyelination, have produced the same results: S1PR-2 knockout or pharmacologic S1PR-2 inhibition reduced MCP and MGL infiltration of the lesions [41]. S1PR-2 is also a negative regulator of oligodendroglial progenitor cell (OPC) maturation into OLGs. Following demyelination, the inhibition of S1PR-2 promoted maturation of OPCs, increased re-myelination and increased the number of mature myelinated axons [41]. In addition to being a receptor for S1P, S1PR-2 has a high affinity for NOGO A, which also inhibits remyelination [54], providing a second S1PR-2 pathway for the inhibition of re-myelination [55,56,57]. Consistent with these observations, the pharmacological inhibition of S1PR-2, or use of S1PR-2 knockout mice, results in improved EAE scores, reduced BBB leakage, less fibrinogen extravasation, reduced macrophage (MPG) and MGL recruitment, as well as increased OLG accumulation and re-myelination [53]. Although not a target of any current clinically employed S1PR modulators, S1PR-2 activation, because of its effects on BBB integrity, macrophage/MGL activation, and myelination, could be an important potential therapeutic target of future drug development.

#### 2.1.3. S1P Neurotropism and S1PR-1/S1PR-2 Synergy

Sphingosine-1-P, by stimulating both S1PR-1 and S1PR-2, may induce gene expression encoding for the production of protective astrocyte neurotrophic factors, not achieved by S1PR-1 modulation alone [58]. Bi-receptor S1PR-1 and S1PR-2 stimulation [59] appears to be necessary to up-regulate neurotrophic mRNA expression, which only takes place in astrocytes [60,61]. To date there are no pharmacological agents of which we are aware that operate jointly or exclusively at both S1PR-1 and S1PR-2 under investigation as potential medications for MS, yet the importance of regulating astrocyte proliferation and astrogliosis would be reasons enough to examine agents that can synergistically modulate both receptor subsets. 

#### 2.1.4. S1PR-5

The S1PR-5 is mainly expressed on OLGs in white matter tracts, and on brain endothelial cells [51,62]. S1P5 receptor mRNA has been found primarily in the white matter of the CNS, but has not been detected in neurons, microglia, or astrocytes [63,64]. S1PR-5 is expressed throughout the development timeline of OLGs, from immature stages to the mature myelin-forming cell [62,65,66]. In mature OLGs, S1P5 receptors co-localize with myelin basic protein exclusively on myelinated axons and not on OLG cell bodies [64]. 

Although S1P5 appears to mediate S1P-induced survival of mature OLGs, it may also induce cell process retraction on immature OLGs [51,62], thus regulating a dual-signaling developmental pathway. It appears that activation of S1PR-5 may be important in the maturation and survival of OLGs by playing a role in their modulation, formation, and myelin repair [62]. 

As has been shown for S1PR-2, S1PR-5 influences BBB integrity and is highly expressed on human brain endothelial cells. Studies have demonstrated that S1PR-5 plays a key role in BBB maintenance and modulation of endothelial cell inflammatory state. S1PR-5 knockdown endothelial cells have reduced barrier integrity [67]. Activation of S1PR-5 reduces the expression of inflammatory cell adhesion molecules on their surface, and enhances the capacity of brain endothelial cells to prevent monocyte penetration. It has been suggested [67] that the role of S1PR-5 may include induction of specific BBB properties such as low paracellular permeability and increased expression of key brain endothelial proteins such as tight junction, ATP binding cassette and glucose transporter molecules. These results point out the potential importance of investigating pharmacological agents which have high affinity and selectivity to promote S1PR-5 preservation in maintaining BBB integrity and protecting OLG maturation and myelination.

### 2.2. Direct Pharmacological Actions of the S1PR Modulators in the CNS

Established and approved S1P agonists include fingolimod (FGM), siponimod (SPM), and ozanimod (OZM). Ponesimod (PNM) has recently been submitted for regulatory approval based upon Phase III clinical study outcomes. Because it has been available the longest, most data on modulation of S1P receptors by these agents come from studies utilizing FGM. At low doses, FGM has been shown to cause process extension of human OPCs via interaction with S1PR-1, but at high doses, causes process retraction via S1PR-3 and S1PR-5 [42]. Furthermore, short-term FGM treatment causes retraction and prevention of migration of OPCs via an S1PR-3 and S1PR-5-mediated pathway, whereas long-term exposure led to increased cell survival via an S1PR-1-driven pathway [42,51]. Fingolimod also enhances BBB integrity, probably by modulation of S1PR-5, reducing transendothelial migration of monocytes in vitro, the latter being an initial step in the formation of new MS lesions [67]. 

#### 2.2.1. Fingolimod 

As noted above, the therapeutic effects of FGM appear to result at least in part from control of lymphocyte trafficking via down-regulation of lymphocyte S1PR-1 (Figure 4) [68,69]. There is also growing evidence from preclinical studies, that some of FGM’s therapeutic effects may be independent of lymphocyte sequestration in peripheral lymphoid organs [38,69]. FGM readily crosses the BBB, potentially enabling its subsequent modulation of S1PRs [35,69,70].

There is up-regulation of S1PR-1- and S1PR-3-expressing astrocytes [71] during active inflammation, and there is growing evidence that astrocytes expressing S1PR-1 may be a primary FGM therapeutic target in the CNS [72]. Consistent with this view has been the demonstration of diminished therapeutic benefit of FGM in EAE in the absence of S1PR-1 expression on astrocytes in genetic knock down mice [38]. 

Fingolimod, which causes the internalization and destruction of astrocytic S1PR-1 [38] also enhances remyelination in a number of different in vitro models and in EAE mice [40,45,73,74,75], prevents OLG death, reduces the number of reactive astrocytes and reduces the number of reactive MGL. These beneficial effects are mimicked by the selective S1PR-1 antagonist CYM5442, which also prevents the up-regulation of S1PR-1 on astrocytes [45]. As further evidence of the importance of the astrocyte S1PR-1 receptor, the S1PR-1 gene is primarily expressed in astrocytes, its coupled-G-protein-regulated pathway promotes astrocyte proliferation, and its selective blockade inhibits astrogliosis, potentially reducing OLG death and demyelination. These results are supporting evidence that at least some of the protective effects of FGM are mediated via S1PR-1 negative modulation in the CNS [45]. As further support, deletion of the astrocyte S1PR-1 gene reduces production of the pro-inflammatory cytokines IL-1β, IL-6 and IL-17, reduces demyelination, and eliminates any additive benefit of CNS protection by FGM treatment [38].

#### 2.2.2. Siponimod

Siponimod is a selective modulator of S1PR-1 and S1PR-5 [76,77], and was originally synthesized to reduce cardiovascular effects by eliminating S1PR-3 affinity [78,79]. Its structure favors avid penetration of the BBB. It has a short T1/2 elimination, and lymphocyte counts may return to baseline within 48 h of the last dose (see Table 2) [78]. Recent studies in both human and rodent astrocytes have demonstrated that SPM modulates cellular pro-survival pathways [80,81,82], primarily via S1PR-1 [82]. SPM also demonstrates anti-inflammatory effects, reducing phosphatidyl choline, TNFα and IL-17-induced IL-6 production by via S1PR-1 modulation. In an in-situ injected curpizone mouse model, SPM treatment reduces myelin breakdown-associated proteins, decreases the number of damaged axons, decreases OLG loss, increases the number of myelinated axons, and reduces MPG and MGL infiltration [83]. In studies of EAE-induced excitotoxic synaptic degeneration, SPM treatment preserves gabaergic neurons in the striatum, and reduces microgliosis, even in the absence of pro-inflammatory lymphocyte affects. These results have been replicated in brain slices, further supporting a direct SPM affect upon the CNS. It has been hypothesized that SPM, via its actions on S1PR-1, reduces MGL activation, which in turn reduces pro-inflammatory lymphocyte recruitment into the CNS [83]. Lastly, since SPM is also a modulator of S1PR-5, preservation of OLGs and myelinated axons in demyelination models following its use is not unexpected. Although, to our knowledge, not yet reported, SPM may also preserve BBB function via S1PR-5 modulation. 

#### 2.2.3. Ozanimod

Ozanimod has a high affinity for S1PR-1 and to a lesser extent for S1PR-5. It selectively crosses the BBB with brain to blood ratio of 10-16:1 [30] and its affinity for the S1PR-1 is comparable to FGM-P and SPM [30]. Like FGM-P and SPM, it induces rapid internalization and degradation of S1PR-1 in rodents and produces reduction in circulating B and T cell lymphocytes [30,84]. Compared to FGM and SPM, there is more rapid lymphocyte reconstitution after it is discontinued. The t1/2 elimination of its major active metabolites, cc112273 and cc1084037, is only 11 days (see Table 2) [85]. Of note, OZM treatment significantly improves EAE scores in mice, even in the presence of restored blood lymphocyte counts [30], supporting a direct CNS therapeutic role. We are unaware of extensive preclinical studies of OZM modulation of S1PR-5, its effects on OPCs, OLGs or myelination, or BBB integrity.

#### 2.2.4. Ponesimod 

Ponesimod is a selective S1PR-1 modulator [86,87], with a t1/2 elimination of 21-33 h [87], which like FGM, SPM and OZM, reduces circulating B and T cell lymphocytes by up to 70%, with return to baseline values within 4 days of cessation (see Table 2) [84,88,89,90], PNM readily crosses the BBB and reduces EAE scores in mice [84].

## 3. Clinical Trial Results

### 3.1. Fingolimod

Fingolimod was the first oral medication, approved in patients with relapsing forms of MS. The therapeutic efficacy of FGM for RMS was initially established in two pivotal, phase III, double-blind, randomized clinical trials of adults, comparing FGM to placebo (FREEDOMS) and comparing FGM to IFN-β-1a (TRANSFORMS). This was followed by the phase III extension studies FREEDOMS II and extended TRANSFORMS. The regulatory approval for use of FGM in the treatment of RRMS in children ages 10–17 years of age in 2018 [91] followed the results of the pivotal trial, PARADIGM [92].

#### 3.1.1. FREEDOMS

FREEDOMS was a phase III double-blind, placebo-controlled trial with 1272 patients that demonstrated superior efficacy of FGM in RMS when compared to placebo [93]. Inclusion criteria included a diagnosis of RMS, Expanded Disability Status Scale (EDSS) score ≤5.5, and evidence of active disease over the previous 2 years. The median disease duration was 6.7 years, and median EDSS at baseline was 2.0. Patients were randomized in a 1:1:1 ratio to receive placebo, FGM 0.5 mg, or 1.25 mg once daily. The primary endpoint was annualized relapse rate (ARR) and the main secondary endpoint was time to three-month confirmed disability progression (CDP). Other secondary endpoints included: time to first relapse, change in EDSS score and the MS Functional Composite (MSFC) z-score at baseline compared to 24-months, conventional MRI measurements, drug tolerability, and safety. Because the 1.25 mg dose of FGM did not demonstrate therapeutic superiority to the 0.5 mg dose, it was the latter that received regulatory approval, and for that reason only the data for the 0.5 mg dose are presented. 

The ARR was 0.18 for 0.5 mg FGM, and 0.40 for placebo-treated patients (*p* < 0.0001), a relative reduction of 54% in favor of FGM. Time to first relapse was significantly longer in FGM than in placebo-treated patients and more FGM patients remained relapse free over 24-months. Cumulative probability of three months of CDP was 17.7% for FGM and 24.1% for placebo (Hazard ratio 0.70). During the 24-month study period, EDSS score and MSFC z-scores remained stable in the FGM-treated group and minimally worsened in the placebo group. FGM treatment was associated with a significant relative decrease in the number of new or enlarging T2 lesions (74%), gadolinium-enhancing (Gd+) lesions (79%) and a relative reduction in brain volume loss (36%), all MRI comparisons being significant at 24 months (*p* < 0.001).

#### 3.1.2. TRANSFORMS

TRANSFORMS was a Phase III clinical trial, enrolling 1292 patients, comparing efficacy of FGM to intramuscular IFNβ-1a in a 12-month, randomized, double-dummy, parallel group study in patients with active RMS over the previous 24 months [94]. The inclusion criteria included a diagnosis of RMS, EDSS ≤ 5.5. Patients were randomized 1:1:1 to oral FGM 0.5 mg or 1.25 mg once daily, or IFNβ-1a 30 μg IM once weekly, and we only report on the FGM 0.5 mg vs. IFNβ-1a results. The primary endpoint was ARR and major secondary endpoints included number of new or enlarging T2 MRI lesions at one year and 3-month CDP during the 12-month duration of the study. At 12 months, ARR was 0.16 for FGM, and 0.33 for IFNβ-1a patients (*p* < 0.001), a relative reduction in ARR of 52%. In the FGM arm time to first relapse was longer and more patients remained relapse free than the IFNβ-1a treated patients. FGM treatment was associated with a relative reduction in the mean number of new and enlarged T2 lesions (FGM 1.7 and IFNβ-1a 2.6; *p* < 0.004), a 54% relative reduction in Gd+ lesions (*p* < 0.001), a 31% relative reduction in percent brain volume loss from baseline (*p* < 0.001). In contrast, there was no significant difference observed between the treatment arms with respect to 3 month CDP.

#### 3.1.3. FREEDOMS II

FREEDOMS II, a phase III trial to evaluate the efficacy and safety of FGM [95] was a 24-month, randomized, double-blind, placebo-controlled, evaluating the efficacy of FGM compared to placebo in 1083 patients. Inclusion criteria included: a diagnosis of RMS, evidence of active disease over the previous 24 months, and an EDSS ≤ 5.5. Patients were randomized in a 1:1:1 ratio to receive FGM 0.5 mg, or 1.25 mg, or placebo once daily but only the FGM 0.5 mg and placebo data are reported; patients randomized to the FGM 1.25 mg arm were switched to 0.5 mg a day at the recommendation of the data and safety committee. The primary endpoint of the trial was ARR. Secondary endpoints included the effect of FGM upon time to 3-month CDP, change in MSFC score, and effect on MRI measurements in comparison to placebo.

The ARR over 24 months was 0.21 for FGM and 0.40 for placebo (*p* < 0.0001), a relative reduction of 48%. The percent change in brain volume from baseline to 24 months, was 0.9% for FGM and 1.3% for placebo, a relative reduction of 31% (*p* < 0.001). The mean number of new and enlarged T2 lesions at 24 months was 2.3 for FGM, and 8.9 for placebo, a relative reduction of 74%; there was a 70% relative reduction in number of Gd+ lesions. There was no significant difference between FGM and placebo in 3- or 6-month CDP. The time to first confirmed relapse was longer, and more patients remained relapse-free in the FGM treated arm compared to placebo-treated patients at 24-months. MSFC scores were also improved at month 24 in the FGM versus the placebo-treated arm.

#### 3.1.4. Extended TRANSFORMS

Extended TRANSFORMS reported long-term results (up to 4.5 years) for the core TRANSFORMS patients [96]. A total of 92% (*n* = 1027) of patients completing TRANSFORMS entered and 75.2% completed the extension phase. Patients randomized to FGM 0.5 mg or FGM 1.25 mg in the core study continued at the same dose in the extension study, whereas patients receiving IFNβ-1a were re-randomized 1:1 to either FGM 0.5 mg or FGM 1.25 mg daily. Following the sponsor’s decision to discontinue FGM 1.25 mg in 2009, all patients subsequently received FGM 0.5 mg until completion of the extension phase. It is only the 0.5 mg FGM results vs. IFNβ-1a which are reported below.

The primary endpoint, ARR at 4.5 years, was significantly reduced in patients who initiated FGM treatment in TRANSFORMS compared to those patients switching from IFNβ-1a to FGM during Extended TRANSFORMS: (0.17 versus 0.27), a 35% relative reduction in risk of relapse (*p* < 0.001). Patients initially treated with IFNβ-1a who switched to FGM, experienced a subsequent 50% reduction in ARR, 0.4 to 0.2, by the end of the extension study. MRI outcomes in the extended TRANSFORMS revealed a 63% reduction in new or newly-enlarged T2 lesion count in the group switching from IFNβ-1a to FGM. The proportion of patients with no evidence of disease activity (NEDA: no relapses, no MRI worsening, and no sustained disability progression) increased by 50% in the first year after switching to FGM (43% to 66%). During the extension phase, there was no statistical difference in the number of patients in either group who remained without Gd+ lesions. The relative reduction in brain volume loss observed during the core study in patients continuously treated with FGM was maintained through the extension study, whereas the patients switched from IFNβ to FGM experienced a reduction in the rate of brain volume loss during the extension phase.

#### 3.1.5. PARADIGM

PARADIGM [92] was a 24 month randomized double-blind, active-controlled, parallel group phase III study of 215 patients with RRMS between the ages of 10 and 17 years of age randomized to either daily oral FGM (107) 0.5 mg (0.25 mg for those weighing 40 kg or less), vs. IFNβ-1a 30 μg IM weekly. All patients had evidence of disease activity in the previous 2 years. The primary outcome, ARR, was 0.12 in FGM versus 0.67 in the IFNβ-1a treated patients, a relative decrease of 82% (*p* < 0.001). The mean percentage of relapse-free patients was 85.7 versus 38.8 favoring the FGM treatment arm. Secondary outcomes included the total number of new and enlarged T2 lesions, which was 4.39 in FGM versus 9.27 in IFNβ-1a treated patients, a 53% relative decrease (*p* < 0.001). The mean number of Gd+ lesions/scan was 0.44 in FGM versus 1.28 in IFNβ-1a treated patients. The mean rate in brain volume change was -0.48% in FGM versus −0.8% in IFNβ-1a treated patients.

### 3.2. Siponimod 

As previously indicated, SPM is a selective S1P1R and S1P5R modulator that readily crosses the BBB [78]. Preclinical studies indicate that it may preserve neurons by preventing excitotoxic synaptic degeneration [83], enhancing cell survival pathways [80,82], and promoting remyelination in the CNS [83]. Given the results of these preclinical studies, and because of the need to develop DMTs that might slow disability worsening in patients with secondary progressive multiple sclerosis (SPMS), it was decided to evaluate SPM efficacy in this patient population. 

#### EXPAND

EXPAND [97] was a double-blind, randomized phase III study of 1645 adults, age 18–60, with SPMS and an EDSS score of 3.0–6.5. Patients were assigned (2:1) to once daily oral SPM 2 mg or placebo for up to three years, or until the occurrence of a pre-specified number of CDP events. The primary endpoint was time to three-month CDP. At baseline, the mean time since first MS symptoms was 16.8 years and time since conversion to SPMS was 3.8 years. Sixty-four percent of patients (*n* = 1055) had not had a relapse in the previous two years; 56% (*n* = 918) needed walking assistance. Twenty-six percent (288/1096) of patients receiving SPM and 32% (173/545) receiving placebo had three-month CDP (hazard ratio 0.79, 95% CI 0.65–0.95), a relative risk reduction of 21% (*p* = 0.013). Of the secondary endpoints, there was no significant difference between SPM and placebo-treated patients in the time to three-month confirmed worsening of at least 20% in Timed 25 Foot Walk (T25FW). The increase in T2 lesion volume from baseline was significantly lower in SPM-treated versus placebo-treated patients, with a between group difference of −695.3 mm^3^ (95% CI –877.3 to –513.3; *p* < 0.0001). Numerically more patients receiving SPM were free of Gd+ lesions (89% vs. 67%) and of new or enlarging T2 lesions (57% vs. 37%) than their placebo-treated counterparts. Siponimod was approved for the treatment of “active forms” of SPMS, defined as patients having had a relapse in the previous 2 years, because this was the only subgroup in which the primary end-point was achieved. Following completion of the core EXPAND trial, an open label extension was initiated which is expected to conclude in 2024. 

### 3.3. Ozanimod

Ozanimod, a S1PR1 and S1PR5 modulator, that has been studied in two phase III clinical trials, RADIANCE Part B and SUNBEAM, the most recent agent in this class to receive regulatory approval. Each of these trials was a randomized, double-blind, double-dummy, parallel group, active-controlled study, evaluating the efficacy and safety of daily oral OZM 0.5 mg or 1 mg versus weekly IFNβ-1a 30 μg IM in patients with RRMS. Following its ingestion, OZM is partly transformed into 2 primary metabolites, CC112273 and CC1084037, which account for the bulk of OZM pharmacological effects, each with a t1/2 elimination of approximately 11 days, compared to 19–22 h for OZM itself [85].

#### 3.3.1. RADIANCE Part B

RADIANCE Part B [98], a 24 month trial, included 1,313 patients with RRMS randomized to daily oral OZM 0.5 mg (*n* = 439), OZM 1 mg (*n* = 433) or weekly IFN β-1a 30 μg IM (*n* = 441). Inclusion criteria were ages 18–55 years, EDSS 0.0–5.0, diagnosis of RRMS, and evidence of active disease over the prior 24 months. Exclusion criteria included recent myocardial infarction, TIA, stroke, prolonged QTc interval, resting heart rate < 55 bpm, Type I diabetes, or uncontrolled Type II diabetes. 

The primary endpoint in RADIANCE Part B was ARR at each OZM dose versus interferon β-1a over 24 months of treatment. Key secondary endpoints included number of new or enlarging T2 brain lesions, number of Gd+ brain lesions, and time to three-month CDP. Other secondary endpoints included relative rate of whole brain volume loss. Exploratory outcomes included relative changes in cortical grey matter and thalamic volumes over the 2 years of observation. Because of its superior efficacy, regulatory approval was only given the 1.0 mg dose. The 0.5 mg data are not presented. ARR at two years was 0.172 for OZM and 0.276 for IFN β-1a, a relative reduction of 38% favoring OZM (*p* < 0.0001). There was a 42% relative reduction in new and enlarging T2 lesions with OZM treatment compared to IFNβ-1a (*p* < 0.001). The relative reduction in number of Gd+ enhancing brain lesions at two years was 53% in OZM-treated patients (*p* = 0.0006) compared to patients treated with IFNβ-1a. There were no statistically significant differences between OZM and IFNβ-1a treatment in time to three-month CDP. Relative reduction in brain volume loss at 2 years was 27% less in OZM (*p* < 0.0001) than in IFNβ-1a treated patients. Relative cortical gray volume loss was reduced by 58% in patients treated with OZM (*p* < 0.0001) and relative thalamic volume loss at 2 years was reduced by 32% (*p* < 0.0001) in patients receiving OZM.

#### 3.3.2. SUNBEAM

The SUNBEAM phase III study [99] included 1,346 patients with RRMS, randomized to daily oral OZM 0.5 mg (*n* = 451) or 1 mg (*n* = 447) versus weekly intramuscular 30 μg IFNβ-1a IM (*n* = 448). Inclusion criteria included ages 18–55 years, EDSS 0.0–5.0, diagnosis of RRMS, with the same remaining inclusion and exclusion criteria as RADIANCE Part B. Only the data for 1 mg OZM is presented.

The primary endpoint in SUNBEAM was ARR with OZM treatment versus IFNβ-1a over 12 months of observation. Key secondary endpoints included number of new or enlarging T2 brain lesions, number of Gd+ brain lesions at one year, and time to three-month CDP. Other secondary endpoints included whole brain volume loss. Exploratory endpoints included cortical grey matter and thalamic volume changes at one year. ARR at one year was 0.181 for OZM and 0.35 for IFNβ -1a treated patients, a relative risk reduction of 48% (*p* < 0.0001) favoring OZM treated patients. There was a 48% relative reduction in the number of new or enlarging T2 brain lesions favoring OZM-treated patients (*p* < 0.001) and total number of Gd+-brain lesions reduced by 63% in the OZM treatment arm (*p* < 0.0001) compared to IFNβ-1a. There was no statistically significant difference between OZM and IFNβ-1a for time to three-month CDP. There was a 33% relative reduction (*p* < 0.0001) in percentage whole brain volume decrease for OZM treated patients, as well as an 84% (*p* < 0.0001) relative reduction in cortical grey matter loss and a 39% (*p* < 0.0001) relative reduction in thalamic volume loss in OZM treated patients. It is of particular interest to learn the results of on-going studies which are focused on cognitive functioning and patient reported outcome status [100] in light of the demonstrated relative reduction in cortical gray matter and thalamic volume loss in OZM treated patients.

The regulatory approval for the higher OZM dose is a 92 mg capsule, as opposed to the 1.0 mg OZM hydrochloride tablets used in the clinical trials [85]. 

### 3.4. Ponesimod

Ponesimod (PNM), a selective S1PR-1 modulator without known effect on the remaining four S1PRs, has been studied in one phase III clinical trial.

#### OPTIMUM

OPTIMUM [101], was a 108 week multicenter, randomized, double-blind, parallel group, and active controlled study, evaluating the efficacy, safety, and tolerability of daily oral PNM 20 mg versus daily oral teriflunomide (TFM) 14 mg in adults with RRMS. OPTIMUM enrolled 1,133 patients, randomized to receive PNM (*n* = 567) or TFM (*n* = 566). Inclusion criteria included ages 18 to 55 years, EDSS 0.0-5.0, diagnosis of RRMS, and active disease in the 24 months prior to screening. The primary end-point was ARR over 108 weeks. Key secondary end-points included change from baseline to week 108 in fatigue-related symptoms, as measured by the symptoms domain of the FSIQ-RMS. Other secondary endpoints included the mean number of combined unique active lesions (CUAL) per year, defined as new Gd+1 lesions and any new or enlarging T2 lesions. In addition, time to 12-week and 24-week CDP was also determined.

There was a 30.5% relative reduction (*p* = 0.0003) in ARR in PNM treated patients (0.202 versus 0.290) compared to TFM treated patients. Ponesimod demonstrated superior improvement in fatigue scores at week 108 compared to TFM as measured by the FSIQ-RMS weekly symptom score (mean difference −3.57, *p* = 0.0019). There was a 56% relative reduction in CUALs per year (1.405 versus 3.164) in patients treated with PNM. No significant differences between PNM and TFM seen for 12-week or 24-week CDP. 

## 4. Safety

### 4.1. Fingolimod

In FREEDOMS, 94% of patients receiving FGM compared to 93% of patients receiving placebo had an AE. The majority of these events were mild to moderate with 7.5% of patients on FGM stopping due to side effects compared to 7.7% on placebo. In TRANSFORMS, 86% of FGM 0.5 mg patients compared to 92% of patients on IFNβ-1a had an AE. However, 6% of patients on FGM versus 4% or patients on IFNβ-1a discontinued due to an AE [93,94]. 

In FREEDOMS, overall, incidence of infections was the same in all treatment groups; however, lower respiratory infections were more common in patients receiving FGM. There were two deaths in the 1.25 mg FGM group, one each of disseminated primary varicella zoster and herpes simplex encephalitis. The overall rate of herpes infections receiving FGM was 0.01 (10/854) and 0.01 (4/418) in placebo treated patients. As of this writing, there have been 36 confirmed cases of PML in greater than 746,700 patient years since FGM approval. In an analysis of 21 PML cases, age at treatment initiation was not a risk factor for the development of PML; however, longer duration of exposure to FGM increased the risk of developing PML [102]. As of February 2019, 46 cases of cryptococcal infections have been reported. (Novartis data on file), the majority of whom were treated for 2 or more years 

Macular edema was slightly more common in patients on 0.5 mg of FGM compared to placebo, 0.5 verses 0.4%, and typically occurred within the first 3 to 4 months of therapy. Because patients with diabetes and uveitis are at higher risk of developing macular edema, they were excluded from the clinical trials. Given the risk of macular edema, a baseline eye exam is recommended within 6 months of starting and then 3 to 4 months after beginning FGM [91]. 

Fingolimod, on average, lowered the heart rate by eight beats per minute. Following the first dose, 0.6% of patients receiving FGM developed symptomatic bradycardia compared to 0.1% in the placebo group, with rare cases of second degree AV block reported. Nonetheless, patients are required to undergo an ECG prior to the first dose observation (FDO), and then at least a 6 h FDO, during which time hourly pulse and blood pressure are obtained, followed by an additional ECG at the end of monitoring. Medications which can cause bradycardia or prolong the QT interval, such as beta blockers. Selective serotonin and norepinephrine reuptake inhibitors are to be used with caution and may prompt, overnight or more prolonged ambulatory cardiac rhythm monitoring. Fingolimod is contraindicated in patients with recent myocardial infarction, unstable angina, stroke, transient ischemic attack, or heart failure within the last 6 months [91]. There was also an average 3 mm Hg increase in systolic blood pressure and 2 mm Hg increase in diastolic pressure in patients on FGM. Elevation in liver enzymes was more commonly seen in the patients receiving FGM, with elevations of three times the upper limit of normal (ULN) or greater were seen in 14% of patients treated with FGM and in 3% of patients on placebo, typically occurring within 6 to 9 months of starting FGM. Peripheral lymphocyte counts dropped by 73% on average in patients receiving FGM, and typically return to baseline values 1–2 after stopping FGM (Novartis data on file). To date, there has been no evidence that lower lymphocyte counts are predictive of increased infection risk [103].

Skin cancer, particularly basal cell carcinoma, may be more common in patients on FGM [91]. Post-marketing studies have also reported rare cases of cutaneous melanoma and squamous cell carcinoma [104,105].

In the pediatric population (PARADIGM), the incidence of AEs was 88.8% in FGM and 95.3% in IFNβ-1a treated patients. The most common side effects in the FGM treated patients were headache (31.8%), viral upper respiratory infections (21.5%), Upper respiratory infections (15.9%), leukopenia (14%), influenza (1.2%), cough (9%), and pyrexia (7.5%). Serious AEs occurred in the FGM treated cohort and 4.7 % of FGM treated patients discontinued the trial because of SAEs. These AEs included convulsions (5.6%), and 3.7% of patients had serious infections, including one case each of appendicitis, cellulitis, oral abscess, and viral pharyngitis. Single SAE cases also included agranulocytosis, arthralgia, auto-immune uveitis, gastrointestinal necrosis, vasculitis, second degree AV conduction block, and small bowel obstruction. 

Fingolimod is not recommended during pregnancy and lactation, based upon its teratogenicity, increased fetal loss and its presence in the milk of lactating animals. Prescribing information currently recommends stopping FGM 3-months prior to planning conception; however this recommendation may require revision in light of recent reports of increased MS breakthrough activity, occurring in 10–25% of patients within 12 weeks of discontinuing FGM [106,107]. Patients with a higher annualized risk of relapse or higher EDSS prior to starting FGM may be at greater risk of rebound disease once stopping FGM [108]. 

### 4.2. Siponimod

The SPM safety profile is similar to that of FGM. In EXPAND [97], 89% of patients receiving SPM and 82% of patients receiving placebo had an AE. Non-serious adverse events leading to discontinuation in the study was 4% versus 3% respectively. Serious adverse events occurred in 18% of the SPM group compared to 15% in the placebo group. 

Rate of infections were also similar in both groups, but upper and lower respiratory infections, as well as herpes zoster reactivation, were more common in the SPM treated group. The risk of herpes infection was 2.5% in SPM versus 0.7% in placebo treated patients. No cases of PML or cryptococcal infections were reported, although these opportunistic infections may emerge as larger populations are treated for longer durations, given the similar mechanism of action compared to FNM. The reduction in the peripheral lymphocyte count was between 20 to 30% which is less than that observed in patients treated with FGM [93,94], but whether this impacts the comparative rate of infection is yet to be determined. 

Liver function abnormalities were observed in 10.7% of SPM treated patients and 3.7% in the placebo group.

Rate of macular edema was 1.8% in patients on SPM and 0.2% of patients treated with placebo. This is higher than 0.4% reported in the phase III clinical trials with FGM [93,94].

Seizures occurred in 1.7% of SPM treated patients and was 0.4% in the placebo group.

The reduction in heart rate was a mean of six beats per minute in the SPM treated group, with the maximum reduction occurring on average 4 h after the first dose. Bradycardia was reported as an AE in 6% of patients receiving SPM compared to 3% receiving placebo. Cases of Mobitz type II or higher degree atrio-ventricular (AV) block were not observed. As a result, first dose observation is not required except for patients with recent history of myocardial infarction, unstable angina, recent stroke, TIA or baseline heart rate of less than 55 beats per minute [109]. Clinicians may also recommend first dose observation for patients on medication which may lower heart rate, particularly those medication that slow AV conduction. 

Elevation in liver enzymes was 10.1% in patients on SPM, versus 3.7% of patients on placebo. Overall, rate of malignancies was the same in each group. 

Medications which inhibit the hepatic enzymes CYP2C9 and CYP3A4 can result in elevated concentration of SPM, and the opposite is true for hepatic enzyme inducers. Furthermore, in patients with the CYP2C9 genotype variants CYP2C9 1*/3* or 2*/3* should receive only 1.0 mg a day maintenance dosing and those with the 3*/3* genotype should not receive SPM at all due to their inhibition of SPM degradation in the liver [109]. 

### 4.3. Ozanimod

In RADIANCE and SUNBEAM, overall AEs and AEs leading to discontinuation were lower in both groups of patients taking OZM compared to interferon beta-1a. Nasopharyngitis, elevated liver enzymes, hypertension, and urinary tract infections were commonly seen in the OZM groups in both clinic trials. No serious opportunistic infections were reported, and the rate of herpes infections were the same across the three treatment groups. 

There was one case of posterior reversible encephalopathy syndrome (PRES) in a patient receiving 1.0 mg OZM, occurring 10 months after starting the medication. Of note, post-approval cases have also been reported in patients treated with FGM [110,111]. Mean absolute lymphocyte count in patients on OZM 1.0 mg were reduced by approximately 45% by 3 months, and maintained at that level thereafter. The rate of macular edema was 0.3% in each OZM and the IFNβ-1a group. Maximum mean reduction in heart rate on day 1 was 0.6 bpm at 5 h in RADIANCE. Four patients treated with OZM had a heart rate less than 45, but had baseline heart rates of 55-64 bpm, and no symptoms were reported. In SUNBEAM, the maximum mean reduction was 1.8 bpm at 5 h, with no rate below 45 bpm reported. No second- or third-degree heart block was observed in either study. Given the minimal reduction in heart rate, FDO is not required for most patients.

As with the other S1PR-1 modulators, mild elevations in liver function enzymes were observed. In RADIANCE, 6.7% of patients on 1.0 mg of OZM had an ALT of 3 X ULN normal, as did 3.9% of patients on IFNβ-1a. In SUNBEAM, the rate was 4.3% for OZM and 2.2% for IFNβ-1a treated patients. 

### 4.4. Contraindications and Cautions

The use of FGM, SPM, and OZM is contraindicated in patients with recent (within the past 6 months) myocardial infarction or stroke, unstable angina, class III/IV heart failure4.4, recent transient ischemic attack and in patients receiving class I and Class III anti-arrhythmics, patients with Mobitz type 2 s or third degree heart block, sino-atrial block, or sick sinus syndrome without a functioning pacemaker. In addition, specifically for OZM there is a contraindication for the concomitant use of monoamine oxidase inhibitors and in patients with untreated severe sleep apnea. The concomitant use of SSRIs, SNRIs and narcotics is not recommended for any of the S1PR modulators [85,91,109]. 

### 4.5. Ponesimod

The proportion of patients experiencing at least one treatment emergent AE (TEAE) in OPTIMUM was similar for both PNM and TFM, although discontinuation due to TEAE was higher in the PMN than the TFM group, 8.7% vs. 6.0% respectively. Premature discontinuations due to elevated liver enzymes and respiratory events were relatively higher in the PNM than TFM groups. The most common AEs of special interest were elevated liver function tests in the PNM group, (22.7% vs. 12.2%), hypertension (10.1% vs. 9.0%), and pulmonary events (8.0% vs. 2.7%). Most ALT increases ≥3X ULN were single transient asymptomatic events that spontaneously resolved on treatment or resulted in protocol mandated discontinuation. Eight seizures were reported in patients treated with PMN compared to one patient on TFM. 

Skin malignancies occurred in five patients treated with PMN and in one patient in the TFM group. Four cases of bradycardia (0.7%) occurred in the PMN 20 mg group and an additional three patients experienced first degree atrioventricular block. None of the events were serious or led to treatment discontinuation. 

## 5. Potential Therapeutic Opportunities 

As is evident from the data presented, the S1PR modulators are an important addition to the list of DMTs for the treatment of MS. In addition to their oral route of administration, an important addition to patient quality of life, with a higher likelihood of compliance and persistence, they offer significant improvement in clinical efficacy at acceptable levels of risk in studies in which they have been compared to “platform” injectable or oral therapies. Important questions and research opportunities remain. Will agents with more potent S1PR-5 modulation improve remyelination, as a result of improved maturation of OPCs and enhanced myelin formation by OLGs? Will their use reduce inflammatory impact of MS by the enhancement of BBB integrity? Will selective S1PR-2 inhibition stimulate remyelination by blocking the NOGO/LOGO membrane complex, and will simultaneous S1PR-1 and S1PR-2 provide greater synergistic regulation of astrocytes than either agent alone? Since S1PR-3 is up-regulated on astrocytes during CNS inflammation [112] can CNS selective S1PR-3 modulators be developed which do not also increase cardiovascular side effects? Lastly, as more is learned about the intracellular pathways activated or inhibited by S1P and its receptors, the development of pharmacological agents which can manipulate these pathways, and the development of agents that can interact with intracellular S1PRs, offer highly specific, potentially fruitful therapeutic opportunities for MS and numerous other diseases, including rheumatoid arthritis [113,114,115], systemic lupus [116,117], polymyotisis [118], ulcerative colitis [119], psoriasis [120], colon cancer [121,122,123], breast cancer [124,125,126], lung cancer [127,128] and atherosclerosis [129,130].

## Figures and Tables

**Figure 1 biomedicines-08-00227-f001:**
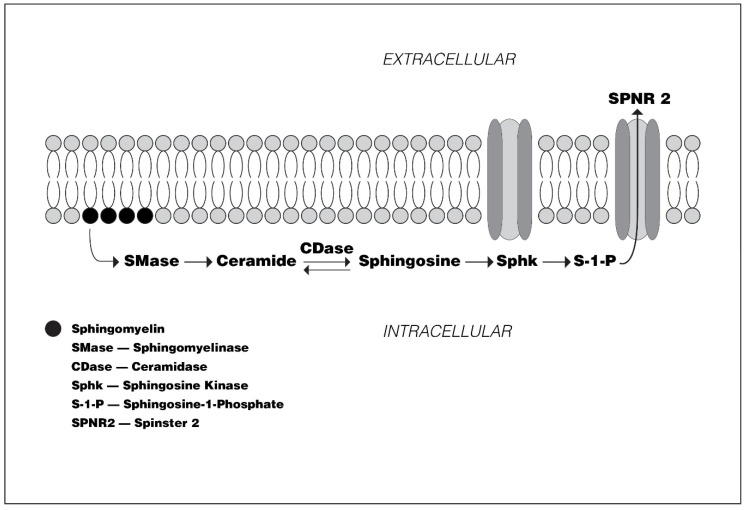
Metabolic Pathway to Sphingosine-1-phosphate Synthesis and Migration from the Intracellular to Extracellular Space via Sphingosine-1-phosphate Transmembrane Transporters.

**Figure 2 biomedicines-08-00227-f002:**
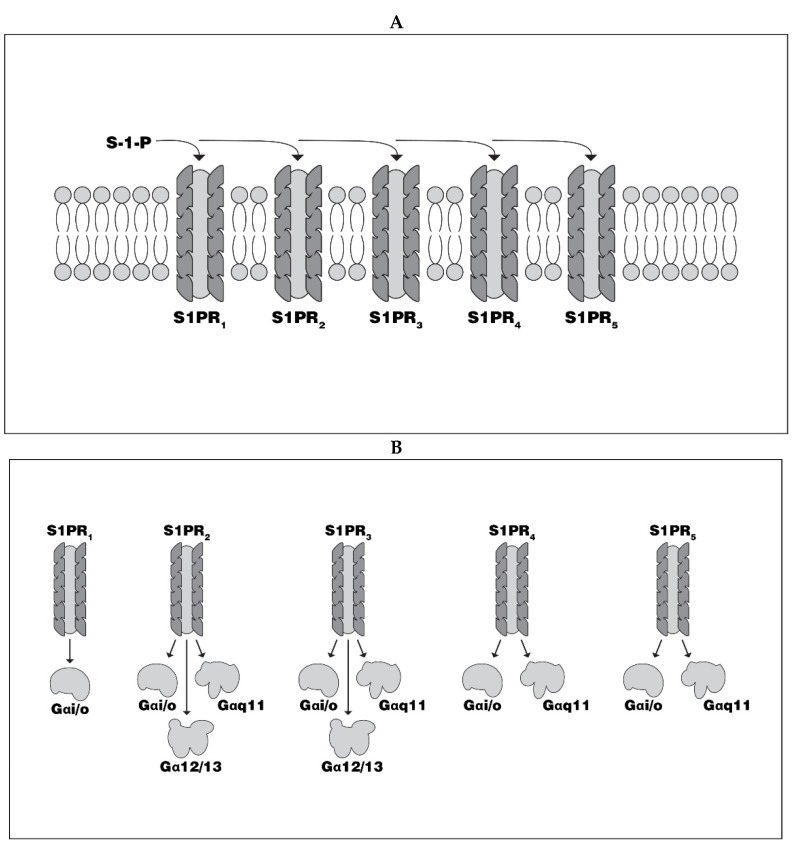
(**A**) Five G protein-coupled Sphingosine-1-phosphate receptors have been identified. (**B**) Each of the S1PRs is coupled to one or more G proteins, resulting in multiple different downstream messaging targets (See Table 1). S-1-P = Sphingosne-1-phosphate (S-1-P), S1PR = Sphingosine-1-phosphate Receptor.

**Figure 3 biomedicines-08-00227-f003:**
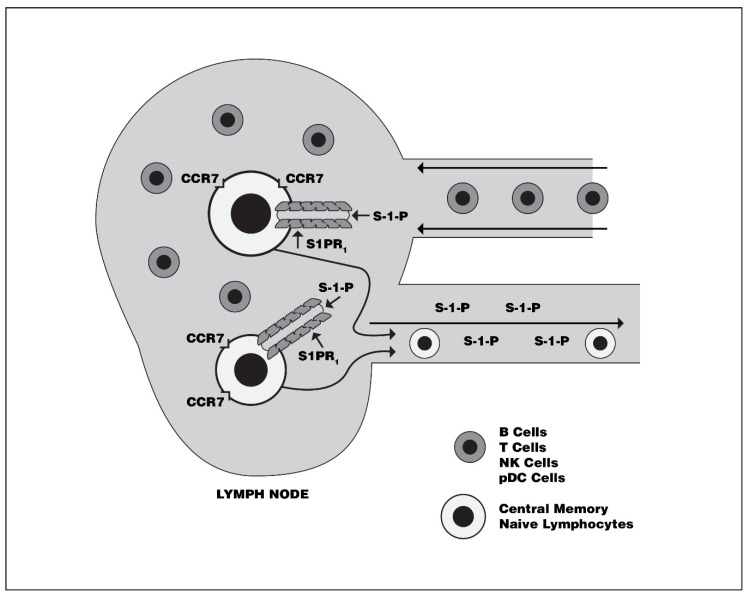
The effect of Sphingosine-1-phosphate on lymphocyte mobilization from peripheral lymphoid organs. Sphingosine-1-phosphate interaction with S1PR1-expressing lymphocytes over-rides CCR7-induced retention signaling, which results in mobilization and egress of naïve and central memory lymphocytes into circulation in response to the S-1-P gradient. Non-CCR7-expressing lymphocytes, such as effector memory cells, are not S-1-P dependent for their entry into the peripheral circulation. B Cells = B cell lymphocytes, T Cells = T cell lymphocyte, NK cells = natural killer cells, pDC Cells = plasmacytoid dendritic cell, S-1-P = Sphingosine-1-phosphate, S1PR1 = Sphingosine-1-Phosphate Receptor, CCR7 = Chemokine Receptor 7.

**Figure 4 biomedicines-08-00227-f004:**
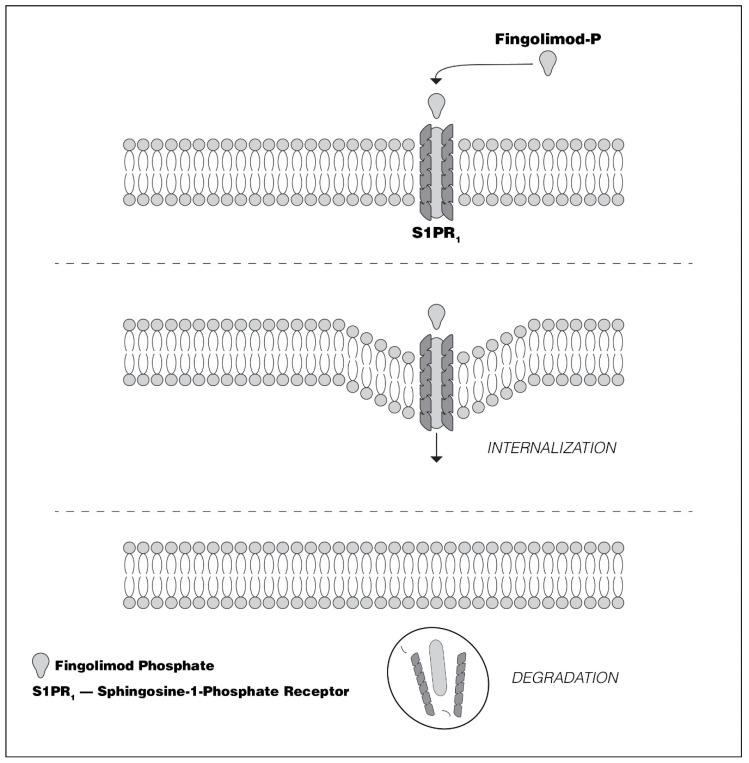
The effect of fingolimod-phosphate upon lymphocyte Sphinogosine-1-phosphate receptor-1. Although initially acting as an agonist, sustained exposure of sphingosine-1-phosphate receptor to fingolimod-phosphate results in receptor internalization and intracellular degradation. As a consequence of sphingosine-phosphate receptor-1 internalization, CCR7-expressing lymphocytes are no longer capable of responding to the sphingosine-1-phhosphate gradient. This results in naïve and central memory cell retention in peripheral lymphoid organs. S1PR_1_ = Sphingosine-1-phosphate receptor-1.

**Table 1 biomedicines-08-00227-t001:** G protein-coupled Sphingosine-1-phosphate receptor subset modulation of downstream signaling pathways.

S1PR	G Protein	Downstream Signaling Pathways
S1PR 1–5	Gαi/o→	Akt→Ras→MAPK→Rac
S1PR 2–5	Gαq11→	DAG→PKC→Ca2+
S1PR 2 and 3	Gα12/13→	Rho→ROCK

S1PR = Sphingosine-1-phosphate Receptor, Akt = serine/threonine kinase (protein kinase B), Ras = small GTPase, MAPK = mitogen-activated protein kinase, Rac = member of Rho family of small GTPases, DAG = Diacyl Glycerol, PKC = protein kinase C, Rho = ras homolog gene family, ROCK = Rho kinase.

**Table 2 biomedicines-08-00227-t002:** Pharmacokinetics and pharmacodynamics of the S1PR antagonists.

S1PR Antagonist	T ½ Elimination	Time to Max Concentration	Median Decrease in Maximum Lymphocyte Count	Maximum Decrease in Steady State Lymphocyte Count	Median recovery Time to Normal Lymphocyte Count
fingolimod	6–9 days *	12–26 h	60% of baseline in 4–6 h	18–30% of baseline	1–2 months
siponimod	30 h	4 h	20–30% of baseline	20–30% of baseline	10 days, but up to 3–4 weeks for some patients
ozanimod CC112273 **	21 h 11 days	6–8 h	30% of baseline	45%	30 days ***
ponesimod	21–33 h	2.5–5 h	Not available	70%	4 days

* Increased by 50% in patients with moderate to severe heart disease, ** CC112273 the major ozanimod active metabolite, *** 90% recovery to baseline lymphocyte count with 3 months.

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
