# Peer review of "Sphingosine-1-Phosphate: Its Pharmacological Regulation and the Treatment of Multiple Sclerosis: A Review Article"

_biomedicines, 2020, doi:10.3390/biomedicines8070227_

Round 1
Reviewer 1 Report
Sphingosine-1-Phosphate pharmacological regulation is important for the treatment of multiple sclerosis. The review summarizes the field. The authors have experience and knowledge in the specific area of the article. The text has both preclinical studies and clinical trial results. However, the material is presented as a list; a graphic plus additional tables would help this article.
Author Response
Dear Reviewer,
The authors appreciate the feedback and agreed that adding graphics and tables will improve the article. Table 1 and 5 figures have been added in the revision to improve the article. Also we will update the article type to review article as this is not a research paper.
Best regards,
Chiayi Chen
Reviewer 2 Report
The authors described the role of S1PR antagonist in promoting lymphocyte sequestration and reduce lymphocyte driven inflammatory damage of the central nervous system in animal models, and they made an overview of the literature to describe the results, obtained in preclinical studies, in models able to mimic multiple sclerosis. Some of these agents have been approved for multiple sclerosis treatment and one has been submitted for regulatory approval. The authors describe Potential Therapeutic Opportunities for selective and intracellular S1PR-driven downstream pathway modulators for multiple sclerosis treatment.
In this reviewer opinion the paper is interesting and need a minor modification before beeing acceptable for publication.
Pag 13 line 587: the authors state “… fruitful therapeutic opportunities for MS and numerous other diseases”. The authors should outline some other diseases they hypotesize.
Author Response
Dear Reviewer,
Thank you for the suggestion to specify the other diseases that share the therapeutic opportunities. We outlined the diseases and provided references in the revision.
Best regard,
Chiayi
Round 2
Reviewer 1 Report
The authors have addressed the concerns in the revised version of the manuscript.